# Unmet Healthcare Needs of Children in Vulnerable Families in South Korea: Finding from the Community Child Center Child Panel Survey

**DOI:** 10.3390/ijerph17218241

**Published:** 2020-11-07

**Authors:** Suyon Baek, Eun-Hi Choi, Jungeun Lee

**Affiliations:** 1Department of Nursing, College of Nursing and Health, Kongju National University, Gongju-si 32588, Korea; whitesy@kongju.ac.kr; 2College of Nursing, Eulji University, Daejeon 34824, Korea; 3College of Nursing, University of Rhode Island, Kingston, RI 02881, USA; jelee@uri.edu

**Keywords:** vulnerable populations, child, unmet healthcare needs, health services accessibility

## Abstract

Presented in this paper is a study that examined the status of unmet healthcare needs of children in vulnerable families and identified factors affecting such unmet needs. The Community Child Center (CCC) Child Panel Survey data in Korea were used. A multiple stepwise logistic regression analysis was performed to examine factors influencing unmet healthcare needs of children. Influencing factors comprised predisposing, enabling, and need factors based on the Andersen Behavioral Model of Health Services Utilization. A total of 340 sixth-graders from vulnerable families participated, and 96 (28.2%) children had unmet healthcare needs. Factors included absence of an after-school caregiver (OR = 1.95, 95% CI [1.16, 3.27]), perceived physical symptoms (OR = 1.33, 95% CI [1.02, 1.73]), parental indifference (OR = 1.33, 95% CI [1.002, 1.77]), duration of daily stay at CCCs (OR = 1.32, 95% CI [1.01, 1.71]), and satisfaction with CCC teachers (OR = 0.65, 95% CI [0.49, 0.85]). The relationship with parents and CCC teachers had the strongest influence on unmet healthcare needs of children. In order to reduce the unmet healthcare needs of children in vulnerable families, existing support structures should be expanded to offer financial and administrative support for children’s parents and CCC teachers.

## 1. Introduction

The national Health Plan 2020 in South Korea, which is used as a foundational element for many national health promotion policies, forecasts that a larger income gap will lead to greater health disparities, causing an increase in medically vulnerable populations [1]. Based on this prospect, the Health Plan 2020 suggests the need to bolster health management for such vulnerable groups [1]. According to the Report for the Current Condition and Policy Issues of Child Poverty [2], approximately 1,034,000 children are categorized as living in families with a household income of less than 60% of the median income in Korea. Furthermore, there were approximately 446,000 single-parent families with children in 2016, which is 8% of all families with children [3].

Children have a limited capacity for choosing their living environment, including family, living situation, school, and community. These environmental factors can have a stronger impact on children’s health than individual factors. Among them, parents’ job situation and household income especially have been identified as important factors [4]. In fact, children in vulnerable families present a higher mortality rate and higher prevalence rates for a variety of physical and psychological problems including obesity, asthma, depression, and anxiety [4,5,6]. 

Despite the prevalent physical and psychological problems experienced by children from vulnerable families, studies on their use of healthcare services and unmet healthcare needs are scarce in Korea [7,8]. Unmet needs, defined as the difference between the services judged necessary to deal with health problems appropriately and the services actually received [9], can prolong the duration of poor health status by worsening the condition and increasing complications [10]. To our knowledge, no previous study in Korea has comprehensively investigated the factors affecting unmet healthcare needs among children from vulnerable families.

The Andersen Behavioral Model of Health Services Utilization (the Andersen model) was developed to define and measure approaches to health services; it suggests that predisposing, enabling, and need factors influence individuals’ use of health services [11]. In this model, predisposing factors mean individual characteristics, including demographic, socio-structural, and attitudinal-belief variables. Enabling factors refer to a condition which allows a family to act on a value or meet a need of health service use, for example income and level of health insurance coverage. Need factors mean physiological and psychological factors related to illness or disability, which directly lead to health service use [12]. The Andersen model has been used to investigate not only health utilization behaviors but also independent variables in studies examining unmet healthcare needs [13,14,15]. Previous studies have used (i) age and education level [13] as enabling factors, (ii) insurance [13,16] and household income [13] as enabling factors, and (iii) urgency [16], mental health need (psychological distress, depression, etc.) [17], subjective health status [13], illness experience [13], and self-rated health [18] as need factors, which were shown to affect unmet healthcare needs. Among these three factors, enabling and need factors affected unmet healthcare needs more strongly than predisposing factors. The Anderson Model has not been applied for Korean children to our knowledge. Therefore, this study examined the current state of unmet healthcare needs among children who visited community child centers (CCCs) based on the Andersen model. 

The Korean government has organized and funded the CCCs since 2004 to facilitate care services for children in the community. In 2018, there were approximately 109,610 children and adolescents who were younger than 18 years old and attended 4211 CCCs. The sizes of the 4211 centers were similar to approximately 69% of the national elementary schools [19]. CCCs in Korea are after-school programs for children from low-income or single-parent families that play an important role as a protective institution for these children [20]. In this study, vulnerable families were defined as families having difficulty in taking care of their children due to low income or being single parent families, and children who attended CCCs were defined as children from vulnerable families. We used data from the Community Child Center Child Panel Survey (the CCC survey) conducted in South Korea, which was designed to provide information for the development of national policies to assist children and adolescents in vulnerable families [21]. The findings of this study will provide evidence to develop policies for reducing unmet healthcare needs for children in vulnerable families.

This study has two specific aims: (1) to describe the status of unmet healthcare needs for children in vulnerable families, and (2) to examine factors influencing these needs. Potential influencing factors include predisposing, enabling, and need factors following the sequence based on the Andersen model [12].

## 2. Materials and Methods

### 2.1. Study Design

Cross-sectional analyses were used to investigate the current state of unmet healthcare needs and related factors among children from vulnerable families based on the Andersen model. The data were obtained from the CCC survey conducted nationwide in South Korea.

### 2.2. Conceptual Framework

The conceptual framework of this study adopted the Andersen model (Figure 1). We included predisposing, enabling, and need factors as factors potentially influencing the unmet healthcare needs of children from vulnerable families based on this model. The variables for each factor were determined based on the available data from the CCC survey.

#### 2.2.1. Andersen Model Factors

Predisposing factors comprise demographic characteristics such as age, gender, and area of residence, social factors such as education and occupation, and mental factors such as health beliefs. Enabling factors comprise financial and organizational factors that enable the use of healthcare services. Financial factors include income and health insurance status, and organizational factors refer to the number of healthcare facilities and providers, locations, and waiting time in healthcare facilities. Need factors include the health status perceived subjectively by patients as well as that evaluated objectively by health professionals. Factors include number of days with limited activity, sick days, number of reported symptoms, and general health status.

#### 2.2.2. Unmet Healthcare Needs

There are several methods to assess unmet healthcare needs, including patient surveys and clinical evaluations by professionals. In this study, unmet healthcare needs refer to a situation where individuals cannot not use the healthcare services they need. This definition has been widely used to assess accessibility of healthcare services in previous studies [22,23].

### 2.3. Data Source and Participants

The CCC survey, commissioned jointly by the South Korean Ministry of Health and Welfare and the Headquarters for CCCs, was conducted annually by the National Youth Policy Institute from 2011 to 2016 [21]. We used the data collected in 2013 and chose a total of 15 study variables based on the Andersen model. There were 35 questions answered by children and eight questions answered by the CCC teachers. Most of the questionnaires included in this survey consist of valid and reliable measurements that have been widely used in Korea. The survey also includes several measurements from the Korean Children and Adolescent Panel Survey, which allows for comparisons between different age groups.

The 2013 CCC Survey data were used in this study, which was the most recent survey since 2011 when unmet healthcare needs were first investigated. The target population included 13,260 children from 3690 centers. The target sample was structured via stratified cluster sampling, and strata were set up based on metropolitan cities and provinces. The strata were chosen based on the number of CCCs and the number of children who attend each CCC across 16 metropolitan cities and provinces. The clusters for the first panel were selected by using the number of average sampling (three or four) in 16 regions of Korea. For example, there were 1141 children attending 367 CCCs in Seoul city, of which 55 children from 18 centers were selected in the first panel, based on the sampling number of three (calculated by 1141/367 = 3). 

The first panel consisted of 638 respondents from 179 centers, and all 638 children completed the survey. Of these respondents, 139 were lost to follow-up in the second survey in 2012, resulting in a total of 499 who completed the second survey. For the third survey in 2013, 150 additional respondents were added, resulting in a total of 515 completing the survey. In 2013, 515 respondents completed the survey. Of the 515 respondents who were sixth graders, 175 respondents were excluded from our analyses because they answered “No” to the question “Have you had any of the following diseases in the past year?” Consequently, a total of 340 respondents were included in the final analysis. The CCC Survey obtained consent from children and parents (legal representatives) before conducting the survey. The consent form included instructions regarding the purpose of the study and the use of the results, the withdrawal of participation in the study if desired, and the anonymization of personally identifiable information. Trained interviewers distributed the questionnaires and provided additional explanations to improve the reliability of the responses for children who had difficulty interpreting questions. The CCC survey data are provided in a de-identified form to those who request it for research purposes. This study only analyzed the survey data provided by the Headquarters for CCCs, and no additional data were collected. 

### 2.4. Ethical Consideration

Participants’ anonymity and confidentiality were maintained, and this study did not pose any threat to them. Furthermore, this study was approved by the institutional review board of the university, to which the authors were affiliated (EU2018-71).

### 2.5. Selection and Definition of Study Variables

#### 2.5.1. Unmet Healthcare Needs

Unmet healthcare needs were assessed via items assessing whether respondents had been treated for any diseases in the area of orthopedics, otolaryngology, ophthalmology, dentistry, internal medicine, pulmonary medicine, dermatology, psychiatry, neurology, or other. Participants who answered “yes” to the question “Have you had any of the following diseases in the past year?” and subsequently responded “I received treatment until it was cured, or I am still receiving treatment” to the question “Have you received treatment?” were classified as having no unmet healthcare needs. Those who answered “I was only partially treated” or “I was not treated at all” were classified as having unmet healthcare needs.

#### 2.5.2. Predisposing Factors

In this study, predisposing factors were gender and area of residence. Regarding area of residence, Seoul and other metropolitan cities were classified as metropolitan areas, and other areas were classified as non-metropolitan areas.

#### 2.5.3. Enabling Factors

Enabling factors in this study were as follows: (1) family health insurance type, (2) living with parents, (3) presence/absence of caregiver after school, (4) degree of conversation with parents, (5) degree of activities with parents, (6) degree of indifference by parents, (7) degree of abuse by parents, (8) duration of daily stay at CCCs, and (9) satisfaction with teachers at CCCs.

Children’s family health insurance type and living with parents were reported by teachers at CCCs. Type of health insurance was classified as Medical Aid and National Health Insurance (NHI). South Korea has NHI, which is a compulsory social insurance system that insures about 97% of the population. The rest of the population is covered by Medical Aid, which is provided for the recipients of the National Basic Livelihood Security System (who have an income of less than 30–50% of the median income). Individuals with Medical Aid need to pay 10–15% of their medical expenses. 

Living with parents was classified as living with “both parents” or with “one parent/other” (for children who lived with only one parent, their grandparents, or other relatives). As for presence/absence of caregiver after school, “present” was assigned when children spent their time after school with a caregiver every day in a week, and “absent” when children spent time after school without a caregiver at least once a week. Regarding degree of conversation with parents, we used the mean scores (out of 5) of the four topic items of discussion including concerns; school life; books, television, or movies; and political or social issues. Scores range from 4 to 20, with higher scores indicating a higher degree of conversation with parents. The Cronbach’s alpha was 0.77 in this study. Degree of activities with parents was measured by using the mean scores (out of 5) of the two activity items including exercise or hobbies and having dinner together. Scores range from 2 to 10, with higher scores indicating a higher degree of activities with parents. The Cronbach’s alpha was 0.53 in this study. Both used a 5-point Likert scale (1 = rarely to 5 = everyday). 

Indifference and abuse by parents were measured using a modified version of the Parenting Behavior Inventory developed by Huh [24]. The mean scores (out of 5) of the three and two items on indifference and abuse, respectively, were used. The three items regarding indifference by parents included “Parents think I am more important than their work,” “Parents ask how I am doing in school,” and “Parents get me proper treatments when I am sick.” A 5-point Likert scale (1 = very much to 5 = not at all) was used. Scores range from 3 to 15, with higher scores indicating a higher degree of indifference by parents. The Cronbach’s alpha was 0.69 in this study. Two items regarding abuse by parents included “Parents scold me a lot when I do something wrong” and “Parents try to hit me first when I do something wrong.” A 5-point Likert scale (1 = not at all to 5 = very much) was used. Scores range from 2 to 10, with higher scores indicating a higher degree of abuse by parents. The Cronbach’s alpha was 0.59 in this study. 

Duration of daily stay at CCCs was measured in minutes from start to end time of CCC attendance. Satisfaction with teachers at CCCs was measured based on the mean score (out of 5) of five items including “Teachers listen to me when I have concerns,” “Teachers are kind,” “I feel comfortable when I interact with teachers,” “Teachers know about my family situation,” and “Teachers know very well how I live.” A 5-point Likert scale (1 = not at all to 5 = very much) was used. Scores range from 5 to 25, with a higher score indicating greater satisfaction with CCC teachers. The Cronbach’s alpha was 0.90 in this study.

#### 2.5.4. Need Factors

The need factors in this study were number of diseases and perceived physical symptoms. Number of diseases was defined as the number of diseases experienced in the past year, including any diseases in the area of orthopedics, otolaryngology, ophthalmology, dentistry, internal medicine, pulmonary medicine, dermatology, psychiatry, neurology, or other. The sum of the scores (1 for having had a disease, 0 for not having had a disease) for the 10 types of diseases was used. A higher score (ranging from 0 to 10) indicated a greater number of diseases experienced in the past year. 

Perceived physical symptoms were measured using a modified version of the scale developed by Jo and Im [25]. The mean score (out of 4) for eight items was used including “I cannot sleep well,” “I often have a headache,” “I often feel nauseous,” “I have a stomachache,” “I do not have an appetite,” “I often feel tired,” “I cannot breathe well,” and “I often feel like I have a fever.” A 4-point Likert scale (1 = not at all to 4 = very much) was used. Scores range from 8 to 32, with a higher score indicating a greater severity of perceived physical symptoms. The Cronbach’s alpha was 0.85 in this study.

### 2.6. Data Analysis

All statistical analyses were performed using Statistical Package for the Social Sciences (SPSS) software version 23.0 (SPSS Inc., Chicago, IL, USA). Participants’ demographic characteristics and unmet healthcare needs were presented as counts and percentages based on a frequency analysis. The study variables measured on Likert scales (e.g., Degree of conversation with parents) were presented using mean and standard deviations. Normal distribution of the study variables was tested using skewness and kurtosis. The study variables including degree of conversation with parents, degree of activities with parents, degree of indifference by parents, degree of abuse by parents, number of diseases, and perceived physical symptoms were normally distributed (skewness < 3 and kurtosis < 8). Children were separated into two groups based on unmet healthcare needs: yes (*n* = 96) and no (*n* = 244).

Independent *t*-tests and chi-square analyses were used to compare differences in predisposing, enabling, and need factors between the two groups. The Benjamini-Hochberg procedure was used to correct for multiple comparisons [26]. The Yates’ continuity correction was used for chi-square analyses to prevent overestimation of statistical significance [27]. Effect sizes were calculated using phi coefficients for chi-squared tests and Cohen’s *d*s for *t*-tests. Factors affecting unmet healthcare needs of children from vulnerable families were identified using multiple stepwise logistic regressions. The study variables were standardized using Z-scores before adding logistic regressions. We tested three models following the sequence: (i) predisposing; (ii) enabling; and (iii) need factors based on the Anderson model. The goodness of fit for the logistic regression model was tested using the Hosmer-Lemeshow test. Two-tailed *p*-values less than 0.05 were considered statistically significant. 

## 3. Results

### 3.1. Participants’ Characteristics and Unmet Healthcare Needs

Table 1 shows the characteristics and state of unmet healthcare needs among children in vulnerable families. Of the 340 participants, 96 (28.2%) had unmet healthcare needs (Table 1). We further examined whether there were any differences between the group without disease and the group with disease. There was no difference in terms of gender, area of residence, type of health insurance, degree of conversation with parents, degree of indifference by parents, degree of abuse by parents, duration of daily stay at CCC, and satisfaction with teachers at CCC. However, 175 children who did not have any diseases in the past year tended to live with parents (χ^2^ = 7.24, *p* = 0.01), have caregivers after school (χ^2^ = 6.09, *p* = 0.02), have fewer activities with parents (t = −2.34, *p* = 0.02), and experienced perceived fewer physical symptoms (t = 4.16, *p* < 0.001), compared to the 340 children who had diseases in the past year (see Appendix A).

### 3.2. Unmet Healthcare Needs by Characteristics of Children in Vulnerable Families

There was no significant difference in unmet healthcare needs according to either predisposing factor: gender (χ^2^ = 0.18, *p* = 0.68) or area of residence (χ^2^ = 0.11, *p* = 0.74). However, there were significant differences in unmet healthcare needs aligned with certain enabling factors: presence/absence of caregiver after school, degree of indifference by parents, duration of daily stay at CCCs, and satisfaction with teachers at CCCs. Unmet healthcare needs were significantly more likely among children without a caregiver after school compared with those who had such a caregiver (χ^2^ = 5.41, *p* = 0.02, phi coefficients = 0.13), among children with a higher degree of indifference by parents (t = −3.17, *p* = 0.002, *d* = −0.38), those with longer duration of daily stay at CCCs (t = −2.12, *p* = 0.04, *d* = −0.25), and those with lower satisfaction with teachers at CCCs (t = 4.05, *p* < 0.001, *d* = 0.48). With regard to need factors, having unmet healthcare needs significantly differed according to perceived physical symptoms, with the severity of perceived physical symptoms being greater among those with unmet healthcare needs (t = −2.69, *p* = 0.01, *d* = −0.32). The Benjamini-Hochberg showed that degree of indifference by parents, satisfaction with teachers at CCC, and perceived physical symptoms were significant (Table 2).

### 3.3. Factors Affecting Unmet Healthcare Needs of Children in Vulnerable Families

The first model (Model 1), tested only with the predisposing factors, showed gender and area of residence were not significantly related to unmet healthcare needs. The Model 2 which added the enabling factors onto the Model 1 showed that unmet healthcare needs were more likely for children who had no caregiver after school compared to those who had such a caregiver (Odds Ratio (OR) = 1.97, 95% confidence interval (CI) (1.17, 3.29)). The likelihood of unmet needs also increased with an increasing degree of indifference by parents (OR = 1.40, 95% CI (1.06, 1.85)), and with lower satisfaction with teachers at CCCs (OR = 0.66, 95% CI (0.51, 0.87)). The final model (Model 3) including all three factors showed unmet healthcare needs were more likely for children who had no caregiver after school compared to those who had such a caregiver (OR = 1.95, 95% CI (1.16, 3.27)). The likelihood of unmet needs also increased with an increasing degree of indifference by parents (OR = 1.33, 95% CI (1.002, 1.77)), with longer duration of daily stay at CCC (OR = 1.32, 95% CI (1.01, 1.71)), with lower satisfaction with teachers at CCCs (OR = 0.65, 95% CI (0.49, 0.85)), and with higher severity of perceived physical symptoms (OR = 1.33, 95% CI (1.02, 1.73)). The goodness of fit of the logistic regression model created to identify the factors influencing unmet healthcare needs was tested using the Hosmer-Lemeshow test. It indicated a good fit, as the fitness index for the final model was not significant (χ^2^ = 11.83, *p* = 0.16) (Table 3).

## 4. Discussion

It has been argued that children’s unmet healthcare needs could be a predictor of poorer overall health in adulthood [28]. Therefore, reducing children’s unmet healthcare needs may not only lower the medical costs associated with health problems in later years but also effectively act as an early intervention to improve adults’ long-term health. Despite such important effects on individuals’ health status, previous studies in Korea have only focused on participants’ current status of healthcare service use and the unmet healthcare needs of children from low-income families and children with disabilities. Consequently, this study is meaningful in that it systematically and comprehensively investigated the factors affecting the unmet healthcare needs of children from vulnerable families based on the Andersen model.

The identified predisposing factors, namely gender and area of residence, did not have a significant effect on unmet healthcare needs. A study on children and adolescents aged between 2 and 17 years found that the predisposing variables, except levels of parent education, were not associated with utilization of care for a dental problem [16]. The authors pointed out that parents may act as a predisposing factor by taking their children to the dentist. Similarly, in our study, unmet healthcare needs were significantly higher for the children with a higher degree of indifference by parents. Conversely, several studies including children [29,30] have found significant associations between predisposing characteristics (e.g., age and racial/ethnic groups) and health care utilization. The participants of these studies included children with diverse racial and ethnic groups who were 0−17 years old, compared to our homogeneous participants of Korean sixth-graders. Further studies on unmet healthcare needs of vulnerable children are therefore needed to examine the more influential predisposing factors.

The type of health insurance, which was chosen as an enabling factor, did not have a significant impact on unmet healthcare needs of children in vulnerable families. However, a study on married immigrant women in Korea found that unmet healthcare needs were greater among non-insured women compared with those with NHI or Medical Aid [14]. A study on American children [31] found that unmet healthcare needs were higher among individuals with health insurance and those non-insured compared with those with private health insurance. Even considering differences between each country’s health insurance system, these results suggest that the type of health insurance has an important impact on unmet healthcare needs, which contradicts our findings. As mentioned earlier, all Korean citizens have either NHI or Medical Aid. Among our participants, 56.2% had Medical Aid, which means that they are responsible for 10–15% of their medical expenses. These characteristics of our participants may appear to influence the results, where the type of health insurance did not affect unmet healthcare needs. However, studies report that Korean health insurance covers 67%, which is lower than 80% of an average coverage of OECD [32], and this leads to unmet healthcare of up to 10% [33]. Therefore, further studies to examine the relationship between health insurance type and unmet healthcare needs of vulnerable children are needed. Beside the inconsistent results, it is important to consider why certain immigrants in Korea do not obtain health insurance. It has been found that many married immigrant women lack knowledge of the application process, rather than eschewing health insurance for financial reasons [34]. With this in mind, countries should implement policies that consider various aspects of the situation of vulnerable populations in their society in order to lower unmet healthcare needs among them. Further investigations that can link policy measures other than the type of insurance to the various causes of unmet healthcare needs among low-income populations are needed [35].

In the present study, the absence of a caregiver after school was found to increase the likelihood of children’s unmet healthcare needs. We could not find any prior studies investigating this relationship. Children have been found to prefer spending time with their parents over economic affluence [36], and the absence of parents in childhood, when spending time with them is highly significant, can lead to depression in adulthood [37]. Therefore, societies need to find a systematic way to reduce the impact of parents’ absence in childhood. It has been suggested that social support lowers the risk of unmet healthcare needs [38]. Thus, if the absence of parents is inevitable in vulnerable families owing to their life situation (e.g., both parents being employed), implementing a social support system that is easily accessible by children will be beneficial.

Degree of activities and conversation with parents were not significant factors for unmet healthcare needs for children in our study, which was different from the significant impacts of the absence of a caregiver after school and parental indifference. A previous study showed inconsistent findings from ours, in that child-parent connectedness affect unmet healthcare needs [39]. Child-parent connectedness was measured by qualitative aspects of relationships with parents using five questions. For instance, children were asked about how close they feel to their parents, and whether they are satisfied with the way they communicate with their parents. In the current study, degree of activities and conversation with parents was measured by quantitative aspects of relationships with parents. Future studies may include different variables to reflect parent-child relationships other than degree of activities and conversation with parents.

In our study, unmet healthcare needs among children in vulnerable families were significantly related to a higher level of parental indifference. This echoes the findings from a previous study on the factors affecting unmet healthcare needs in 18,924 adolescents, indicating that connectedness between adolescents and their parents had a significant effect on unmet healthcare needs [39]. Indifference means not having the proper physical, environmental, medical, cognitive, or emotional care or supervision that a child should receive [40], which may lead to unmet healthcare needs. Parental indifference has been referred to as “silent violence” and found to have a detrimental effect on children’s growth and health [41]. According to a recent report by the National Child Protection Agency in Korea, 33.6% of indifferent parents lacked attitudes and knowledge regarding appropriate parenting, and 22.8% experienced stress related to their socioeconomic status and social isolation [42]. Therefore, indifference could be prevented by implementing education programs regarding attitudes and knowledge for parenting as well as providing childcare services for vulnerable families. Although CCCs and case management services are currently provided for vulnerable families in Korea to protect children at risk, there is an issue regarding insufficient resources and accessibility.

Meanwhile, abuse was not a statistically significant factor of unmet healthcare needs in our study. This may suggest that parents’ indifference and lack of caregiving have a stronger impact than verbal, physical, and emotional abuse on unmet healthcare needs of children in vulnerable families. However, abuse is clinically significant as shown in a recent study where one of the factors associated with caseworker referral to mental health services for maltreated children included abuse history [43]. In addition, abuse was measured based on two items in the present study, and its apparent insignificance may have resulted from problems with this assessment. Furthermore, indifference by parents was measured using three items in the current study. The five items included in measuring parental indifference and abuse were chosen from the valid and reliable instrument [24]. However, the original instrument was developed for use with adolescents and contained 11 items for indifference and abuse. Thus, these five items may not reflect development and environmental aspects of children who use CCCs, and that may affect our results. Future studies need to explore the impact of indifference and abuse on unmet healthcare needs, using more appropriate instruments designed for vulnerable children.

Duration of daily stay at CCC, which was set as an enabling factor, had a significant impact on unmet healthcare needs, with unmet healthcare needs being more likely for children with longer stays at CCCs. This is in line with a previous study suggesting that unmet healthcare needs increase when both parents work, as their work hours are likely to overlap with the operating hours of available healthcare facilities [22]. In other words, parents in vulnerable families cannot spend enough time with their children or take them to healthcare facilities or hospitals owing to their socioeconomic situation, which is in line with our findings. Thus, we suggest paying more attention to the functions of CCCs, which could serve as a protective factor reducing the unmet healthcare needs of children in vulnerable families, since such needs were greater among children who spend more time at CCCs. 

Our findings demonstrated that children who were more satisfied with their teachers at CCCs were less likely to have unmet healthcare needs than those who were less satisfied. The effect size of this factor was small, but the largest among all significant factors. Teachers at these centers take responsibility for and manage children’s after-school hours, spend time with them until their parents return, and protect, guide, and supervise children in place of their parents or primary caregivers [44]. It has been suggested that CCC teachers are key actors who not only provide academic and emotional support as educators and protecting figures, but can also improve the health status of vulnerable children [45]. Our results confirm that CCC teachers can play a protective role in reducing the unmet healthcare needs of these children. However, adequate institutional support is lacking for CCC teachers to spend enough time to provide quality care for vulnerable children in Korea. It has been shown that low salaries and heavy workload lead to a high rate of CCC teacher attrition. In addition, most of their work comprises administrative tasks (68.8%) as well as meal services and cleaning (55.4%) [6]. Based on the largest effect size of satisfaction with CCC teachers for children’s unmet healthcare use, financial and administrative support is needed so that CCC teachers can detect children’s unmet health care needs by spending more time with them, paying close attention, and building nurturing relationships.

The number of diseases was not significantly associated with unmet healthcare needs, which is inconsistent from the results of previous studies among children [43] and older adults [46]. In the present study, children stated the number of diseases based on their memory. On the contrary, caregivers reported clinically significant problems of children through a checklist in the previous study [43]. Thus, the different method of measuring the number of diseases for children may affect our results. In addition, older adults living alone reported they have an average of three chronic conditions, which affected their unmet healthcare needs significantly [46]. Participants in the current study are children who were 12 years old and had an average of 1.5 diseases in the previous year. Additionally, there was no significant difference in the number of diseases between a group with unmet healthcare needs and a group without unmet healthcare needs. Therefore, it is necessary to carefully select and investigate the variables constituting the need factors when applying Andersen’s model in research on children. 

Finally, the likelihood of unmet healthcare needs increased with the severity of perceived physical symptoms, which was set as a need factor. This is similar to previous results showing that unmet healthcare needs among the general adult population increased with poorer subjective health conditions [13]. These findings suggest that when individuals perceive their physical symptoms to be severe, there is a potential increase in unmet healthcare needs. Prior research [14,17] on unmet healthcare needs has typically investigated the impact of subjective health status as a need factor measured on a scale from “good” to “poor.” In contrast, the variable used in this study was “perceived physical symptoms,” which measured specific and multiple items, instead of using the somewhat vague “recent health status” ranging from “I am frequently sick” to “I am very healthy.” As perceived physical symptoms were found to have a significant impact on unmet healthcare needs in our study, we propose that subsequent research on the health status of children may benefit from using specific items that consider various aspects of health as opposed to a single, vaguely phrased item.

In Korea, starting from 2020, a new national healthcare service will be provided to older adults with mobility problems who do not have any family members or caregivers to assist them in using health services. This service will provide actual support that can lead to reducing unmet healthcare needs among elderly population [47]. A similar national healthcare service should be provided for children from vulnerable families who do not have parents or caregivers to assist them in using health services. For instance, CCC teachers can help them. Furthermore, the development of guidelines regarding how to assess the health status of children through their caregivers will help to lower unmet healthcare needs among children in vulnerable families [48].

The limitations of this study are as follows. First, we used secondary data with a limited number of variables to choose from for performing regression analysis. Further studies are needed to identify factors affecting unmet healthcare needs extensively by adding variables such as maternal exposure to violence [49], parental psychological distress [50], and parental substance abuse [51]. Second, reliability coefficients of measurements showed some of the measurements used in the study were inadequate. The measurements with the Cronbach’s alpha lower than 0.70 included degree of activities with parents, and indifference and abuse by parents. Future studies should include more reliable and valid measurements for those variables. In addition, the measurements included in the CCC survey could be modified based on inadequate reliability coefficients. Third, the analyses were based on a cross-sectional design, thus the results of this study cannot prove causal relationships between influencing factors and unmet healthcare needs for children in vulnerable families. Reciprocal associations between a few factors (such as satisfaction with CCC teachers and perceived physical symptoms) and unmet healthcare needs are plausible. Longitudinal studies are warranted to examine the causality of these relationships. Fourth, most of the items in the CCC survey were self-reported. Not all children are able to understand the survey questions and engage in memory retrieval, thus their cognitive capabilities should be considered. Furthermore, self-reported answers may be underreported or exaggerated. Although trained interviewers helped children complete the questionnaires, self-reported answers might affect our results. Further studies with additional reports by teachers and parents are needed to better understand factors affecting unmet healthcare needs for vulnerable children [52].

## 5. Conclusions

This study found that absence of a caregiver after school, indifference by parents, duration of daily stay at CCCs, satisfaction with teachers at CCCs, and perceived physical symptoms had significant effects on unmet healthcare needs of children in vulnerable families. Type of health insurance, which was a significant factor affecting unmet healthcare needs in previous studies, did not demonstrate a significant effect in this study, and parental abuse was not a significant factor either. The relationship with both parents (i.e., absence of caregiver after school, parental indifference) and teachers at CCCs were found to have the strongest influence on unmet healthcare needs among children from vulnerable families. Thus, to reduce unmet healthcare needs of such children, support, which has been largely focused on health insurance, needs be expanded to offer financial and administrative support for children’s parents and CCC teachers.

## Figures and Tables

**Figure 1 ijerph-17-08241-f001:**
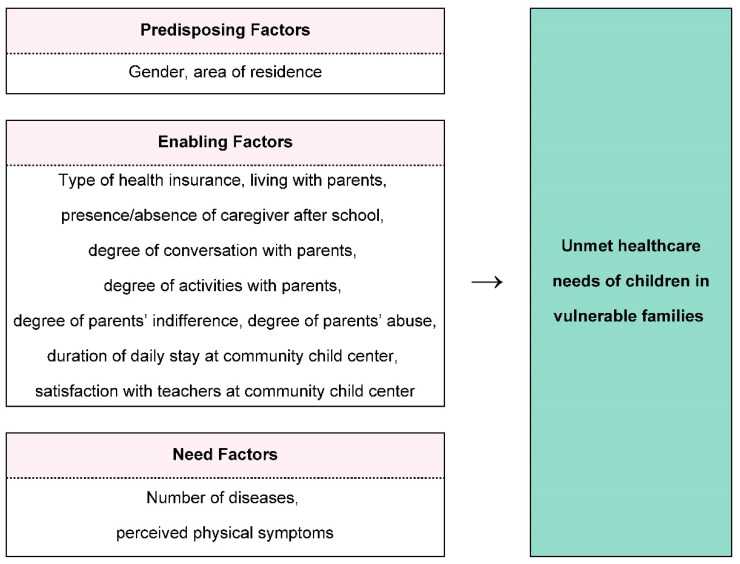
Conceptual framework (Source: the Authors).

**Table 1 ijerph-17-08241-t001:** Participants’ characteristics and unmet healthcare needs (*N* = 340).

Characteristics		Category	*n (%)*	Total Score (*M* ± *SD*)	Item Score (*M* ± *SD*)
Predisposing factors	Gender	Boys	146 (42.9)		
		Girls	194 (57.1)		
	Area of residence	Metropolitan region	128 (37.6)		
		Non-metropolitan region	212 (62.4)		
Enabling factors	Type of health insurance	National Health Insurance	149 (43.8)		
		Medical Aid	191 (56.2)		
	Living with parents	Both parents	224 (65.9)		
		One parent/other	116 (34.1)		
	Caregiver after school	Present	167 (49.1)		
		Absent (one day or more/week)	173 (50.9)		
	Degree of conversation with parents			8.2 ± 3.5	2.1 ± 0.9
	Degree of activities with parents			5.6 ± 2.3	2.8 ± 1.1
	Degree of indifference by parents			6.3 ± 2.5	2.1 ± 0.8
	Degree of abuse by parents			5.2 ± 2.0	2.6 ± 1.0
	Duration of daily stay at CCC (min)			174.9 ± 75.1	174.9 ± 75.1
	Satisfaction with teachers at CCC			18.5 ± 4.3	3.7 ± 0.9
Need factors	Number of diseases			1.6 ± 1.0	1.6 ± 1.0
	Perceived physical symptoms			15.0 ± 4.8	1.9 ± 0.6
Unmet healthcare needs		Yes	96 (28.2)		
		No	244 (71.8)		

*Note.* CCC: community child center; M: mean; SD: standard deviation.

**Table 2 ijerph-17-08241-t002:** Differences in unmet healthcare needs by characteristics of children in vulnerable families (*N* = 340).

Factor	Characteristics	Category	Unmet Healthcare Needs		χ^2^ or t	*p*	Φ (*p*)/ Cohen’s *d*
			Yes	No			
Predisposing factors	Gender	Boys	39 (26.7)	107 (73.3)	0.18	0.68	0.03 (0.59)
		Girls	57 (29.4)	137 (70.6)			
	Area of residence	Metropolitan region	38 (29.7)	90 (70.3)	0.11	0.74	−0.03 (0.64)
		Non-metropolitan region	58 (27.4)	154 (72.6)			
Enabling factors	Type of health insurance	National Health Insurance	47 (31.5)	102 (68.5)	1.16	0.28	0.07 (0.23)
		Medical Aid	49 (25.7)	142 (74.3)			
	Living with parents	Both parents	65 (29.0)	159 (71.0)	0.1	0.75	−0.02 (0.66)
		One parent/other	31 (26.7)	85 (73.3)			
	Caregiver after school	Present	37 (22.2)	130 (77.8)	5.41	0.02	0.13 (0.01)
		Absent (one day or more/week)	59 (34.1)	114 (65.9)			
	Degree of conversation with parents		2.1 ± 0.8	2.0 ± 0.1	−0.28	0.78	−0.03
	Degree of activities with parents		2.8 ± 1.2	2.8 ± 1.1	0.97	0.84	0.02
	Degree of indifference by parents		2.3 ± 0.8	2.0 ± 0.8	−3.17	0.002 *	−0.38
	Degree of abuse by parents		2.7 ± 1.0	2.6 ± 1.0	−1.13	0.26	−0.14
	Duration of daily stay at CCC (min)		188.6 ± 86.1	169.6 ± 69.7	−2.12	0.04	−0.25
	Satisfaction with teachers at CCC		3.4 ± 0.8	3.8 ± 0.8	4.05	<0.001 *	0.48
Need factors	Number of diseases		1.5 ± 0.9	1.6 ± 1.0	0.73	0.47	0.09
	Perceived physical symptoms		2.0 ± 0.6	1.8 ± 0.6	−2.69	0.01 *	−0.32

*Note.* CCC: community child center; **χ^2^**: Yates’ continuity correction; * Benjamini-Hochberg *p* value < 0.05; Φ: Phi coefficients.

**Table 3 ijerph-17-08241-t003:** Factors affecting unmet healthcare needs of children in vulnerable families (*N* = 340).

Factor	Characteristics	Category	Model 1	Model 2	Model 3
			OR (95% CI)	OR (95% CI)	OR (95% CI)
Predisposing factors	Gender	Boys (ref.)	1	1	1
		Girls	1.13 (0.70, 1.83)	1.24 (0.73, 2.10)	1.15 (0.67, 1.98)
	Area of residence	Metropolitan region (ref.)	1	1	1
		Non-metropolitan region	0.90 (0.55, 1.46)	0.98 (0.57, 1.68)	0.95 (0.55, 1.64)
Enabling factors	Type of health insurance	National Health Insurance (ref.)		1	1
		Medical Aid		0.86 (0.51, 1.47)	0.88 (0.51, 1.51)
	Living with parents	Both parents (ref.)		1	1
		One parent/other		0.93 (0.53, 1.66)	0.92 (0.52, 1.65)
	Caregiver after school	Present (ref.)		1	1
		Absent (one day or more/week)		1.97 (1.17, 3.29) *	1.95 (1.16, 3.27) *
	Degree of conversation with parents			1.15 (0.84, 1.56)	1.12 (0.82, 1.54)
	Degree of activities with parents			1.00 (0.74, 1.34)	1.02 (0.76, 1.38)
	Degree of indifference by parents			1.40 (1.06, 1.85) *	1.33 (1.002, 1.77) *
	Degree of abuse by parents			1.17 (0.90, 1.52)	1.13 (0.87, 1.47)
	Duration of daily stay at CCC (min)			1.28 (0.99, 1.65)	1.32 (1.01, 1.71) *
	Satisfaction with teachers at CCC			0.66 (0.51, 0.87) *	0.65 (0.49, 0.85) *
Need factors	Number of diseases				0.83 (0.64, 1.08)
	Perceived physical symptoms				1.33 (1.02, 1.73) *

*Note.* OR: odds ratio; CI: confidence interval; CCC: community child center; *: *p* < 0.05.

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
