# Peer review of "Unmet Healthcare Needs of Children in Vulnerable Families in South Korea: Finding from the Community Child Center Child Panel Survey"

_ijerph, 2020, doi:10.3390/ijerph17218241_

Round 1

Reviewer 1 Report

Abstract

The abstract is written well, but could be made more concise by only using well-known abbreviations for common statistics (e.g., OR, CI) without first defining them, consistent with APA and AMA formatting. Similarly, ORs should be rounded up to the hundredths place (e.g., OR 1.95 vs 1.945). The factors should likely be reported in their order of magnitude (i.e., largest to smallest effect sizes).

Introduction

The authors do an excellent, concise job of describing the significance of their study, as well as how their study is novel and fills a gap in the literature that is consistent with IJERPH’s aims and scope. In some cases, excellent, timely definitions are provided (e.g., unmet needs). However, in other cases, constructs need to be better defined, such as how the Anderson model defines and distinguishes “predisposing”, “enabling”, and “need factors”. This is especially crucial given those three constructs are central to the study methods and results. Relatedly, the introduction should include some of the results from prior studies using the Anderson models (e.g., are certain types of factors more prevalent or predictive or unmet needs?). In some cases, this information is provided in subsequent sections, but this content should be moved up to the Introduction. This would also make the Methods section more coherent in semantic flow.

Additionally, a little more information on South Korea’s CCCs is needed–especially for foreign readers that are likely unfamiliar with South Korea’s CCCs. For example, are CCCs funded and/or staffed by government or private capital and staff? What ages do they serve? How common, numerous, and/or large are they?

Overall, the aims of the study are defined and justified. However, a more thorough literature review of the trichotomous Anderson model (see above) might suggest that certain directional or specific hypotheses are warranted. Otherwise, the aims remain solely exploratory versus using the hypothetical-deductive aspect of the scientific method.

Methods

The use of stratified cluster sampling is excellent. Presumably, the strata were based on the population size versus numbers of cities and provinces. However, this should be made explicitly clear.

The authors initially state the survey’s target population was 13,260 children. They then mention a “first panel” of 638 respondents, but no mention of other panels are made. This needs to be clarified. Similarly, it is unclear whether surveys were administered to 638 potential respondents or 638 people responded to the survey. This is important for understanding the relative response rate of the survey, the representative of the analyzed sample, and the generalizability of the results. Similarly, did the 175 respondents who did not complete the unmet healthcare needs items differ on any CCC Survey item from those who did complete the unmet health needs items? These t-tests and/or chi-square tests should be run and reported, once again to demonstrate appropriate external validity of results and related implications (as their case-wise deletion trims approximately a third of all panel 1 respondents).

The CCC Survey includes several questionnaires, which are detailed in response options and items. However, the psychometrics of these questionnaires and items remains unestablished. If these questionnaires have never been used before, that should be explicitly stated, and at the very least, internal consistency for scales should be provided. Relatedly, why were mean scores used versus total scale scores?

In child development and traumatology, “neglect” is a construct that usually connotates a certain level of severity consistent with other forms of child maltreatment. The items used in this study do not seem to adequately or clearly assess this construct. For example, the item “parents get me proper treatments when I am sick” seems to be a face-valid item to measure medical neglect.  Notwithstanding, both scholarly and legal fields disagree on whether medical neglect should be deemed neglect if it occurs due to familial poverty or lack of sufficient fiscal resources–something which this item does not address. In contrast, the other two items seem focused on emotional neglect, or more accurately, on parenting style constructs such as involvement and responsiveness–although those constructs also seem to overlap with the “degree of conversation with parents” and “degree of activities with parents” items. The authors might consider using a more accurate label or theoretically grounded organization of items. They should also attenuate their discussion of their results to more accurate report that a lower level of engagement and/or responsiveness predicted unmet health needs (since the authors used the entire dimensional variables versus cutoff scores that signified neglect). These issues are exacerbated by the aforementioned lack of reported psychometrics for the scales.

These same issues pertain to a lesser degree to the “abuse” questionnaire, results, and discussion, though the authors do explicitly note the limits of using an unvalidated pair of items.

The Methods section report the authors used multiple stepwise logistic regressions; whereas, the abstract reports they used a multivariate logistic regression. Multivariate regression denotes multiple outcome variables (or variates), while multiple regression denotes multiple predictor variables. Please rectify accordingly.

Given the large number of t-tests and chi-square analyses, some familywise-error correction method (e.g., Bonferroni, false discovery rate correction) should be used to address the inflated Type I error rate.

Results

For Tables, titles should be in title-case. Also, Latin-based statistical symbols (i.e., N, M, SD) should be italicized.  

Other than means and standard deviations (which should only be reported to the tenths place), statistical results should be reported only to the hundredths place with a space placed before and after equal signs (e.g., p = .63 versus p=.627).

Effect sizes should be reported for all statistical tests. For example, Cohen’s ds (or Hedges’ gs) should be reported for all of the t-tests conducted, especially those listed in Table 2. For chi-squared tests, please use an appropriate standardized effect size, such as phi coefficients or Cramer’s Vs (the former is more appropriate for 2x2 chi-square tests). Standardized effect sizes allow readers to make direct interpretative comparisons both within and across studies. Odds ratios are typically unstandardized (unless the variables are first standardized to z-scores, in which case, the authors would need to report this statistical correction). Without standardized effect sizes, the authors (much less the readers) cannot appropriately consider the practical magnitude of study results and related implications.

The authors need to do a better job of justifying why they nested the three models in the reported sequence. This should be more clearly explained in the Methods, but also justified or foreshadowed with supporting literature and/or hypotheses in the Introduction. Namely, the authors need to explain why they first tested predisposing factors, versus first testing enabling or need factors. This sequence may relate to the Anderson model, but if so, no theoretical, conceptual, or statistical reasons or rationale are provided.  

Discussion

The authors point out that analyzed predisposing factors (i.e., child gender and area of residence) were nonsignificant predictors, in contrast to prior studies that used the Anderson model. How many studies have used the Anderson model with youth–and particularly with youth similar in age to this study’s sampled youth? Relatedly, the authors should expand this section to discuss or suggest potential reasons why this discrepancy of results occurred. For example, did this and past studies use markedly different measures of these constructs, or perhaps developmental periods might moderate the relation between these predisposing factors and unmet health needs (especially since youth are dependent upon caregivers for health needs being met). The same level of interpretive analysis should be applied to the other nonsignificant predictors (e.g., health insurance type).

Once the authors compute and report standardized effect sizes, they can then expand their discussion to discuss the relative magnitude of significant effects. This could help highlight which types of interventions or intervention targets might best ameliorate unmet health needs.

The authors correctly note that one limit of the CCC survey (and particularly the measurement of unmet health needs) is that it relied on self-report. The authors also note that “they may be biased”, but they do not discuss in what way, or how that bias may have affected their results. Greater elaboration is needed here for readers to more accurately consider the validity and implications of the study’s findings.

Miscellaneous

Traditionally, paragraphs have 3 or more sentences. Several paragraphs in this paper only have 2 sentences. Consequently, some reorganization of semantic content could be done, to either further develop each topic or combine multiple subtopics into one.

Ranges should use en dashes versus hyphens (e.g., line 51: [12–14] versus [12-14]).

Line 189: Should be “means and standard deviations” versus the singular forms.

For subheadings, please use consistent title-case (e.g., Line 222: Factors Affecting Unmet Healthcare Needs of Children in Vulnerable Families, versus Factors affecting unmet Healthcare Needs of Children in Vulnerable Families).

Author Response

I would like to extend my deep appreciation to you for review.

I thankfully found this round of editing a good chance of improving the paper.

Reviewer 2 Report

The article " Unmet healthcare needs of children in vulnerable families in South Korea: findings from the Community Child Center Child Panel Survey" by Baek S, Choi EH, and Lee J, aims to describe the status of unmet healthcare needs for children in vulnerable families and to examine factors influencing these needs. 

This work is interesting and in my opinion can be considered for publication but the following points need to be addressed

  1. A better description for "study design": described at page 2 as secondary data analysis to investigate the current state of unmet hc needs and factors affecting, while described as "cross-sectional design" at page 10.
  2. Population sampling: 13.260 (target population) from 3690 centers, leading to 340 respondents in the final analysis, is the sample representative? 
  3. Difficulty for statistical support of discussion and conclusions leading to a difficult causal correlation between influencing factors and unmet healthcare needs (rightly cited as a limit, pg. 10).
  4. Authors should be more careful in the discussione and conclusion section
    due to the low number of respondents, with around a 34% that did not
    respond,maybe leading to an important bias.

Author Response

(The authors gave the same response as above.)

Round 2

Reviewer 1 Report

Abstract

The inclusion of effect sizes and specific factors significantly improves the abstract. However, there are some minor formatting errors. For example, ORs should be followed by an equal sign, and LL and UL ORs should be in brackets and divided by a comma versus a hyphen, like so: (OR = 1.95, 95% CI [1.16, 3.27]). This correction should be applied throughout the document.

Introduction

The added material to the Introduction significantly improves the manuscript. The authors should place a paragraph break between lines 61 and 62, since Lines 47–61 focus on describing the Andersen model; whereas, lines 62–74. One other small addition would be adding in a transitional sentence or clause at the end of the suggested paragraph about the model, or at the beginning of the subsequent one. For example, the authors might note explicitly how the current study fills a gap in the research literature (e.g., if the Andersen model has been used in Korean samples, but perhaps not for Korean children in CCCs).

Methods

Once again, the added details strengthen the paper, particularly the sampling methods. For the Cronbach’s alphas, the Greek alpha symbol should not be italicized (unlike Latin-based statistical notations). Moreover, the authors need to recognize how several of these measures had inadequate reliability coefficients. Namely, a value lower than .70 is considered inadequate (though see Taber [2017] for a more nuanced review of such labeled ranges), but the study’s measures of reported caregiver activity with children (.53) and caregiver abuse (.59) call into question not only the reliability of the items (specifically the degree to which individual items were measuring the same construct), but also related validity of data based on said study-created scales. At the very least, this should be noted as a limitation of the study and the rigor of its results and related interpretations (as well as highlighting future areas of research).

The authors should provide a citation for the Benjamini-Hochberg and Yates’ corrections, since these are not readily known family-wise error correction methods. Also note on Line 225, it should read Cohen’s ds for t-tests (i.e., italicize Latin-based statistical notation). Please correct the results section as well. On Line 230, it should read p-values versus P-values.

Results

The inclusion of standardized effect sizes greatly improves the study, particularly as the individual effect sizes are non-trivial (though small). Similarly, Cohen’s ds may be reported simply as ds without repeatedly calling them Cohen’s ds (e.g., Lines 253–254).

Line 271, there appears to be a missing “and” before “with lower satisfaction...”.

Line 282, there should be a space before and after the “=” sign.

For Table 3, the authors may want to note the statistically significant predictors/ORs with an asterisk (with an accompanying note below the table).

Discussion

When noting the non-significance of predisposing factors for this study, the authors briefly note how past studies have found divergent results. However, the authors need to better discuss why they think these disparities exist. For example, are there significant differences in those studies’ samples, sampling methods, measures of predisposing factors and/or outcome variables, statistical power and methods, etc., that might explain the different empirical results? What does it say about the empirical validity of the theoretical model? Relatedly, the authors could then note how future studies might clarify these issues/problems. This level of greater analysis and prescription is still needed for most of the manuscript (e.g., Lines 335–340).

Line 381: Technically, a Cohen’s d of 0.48 is still small (0.50–0.79 is medium).

Author Response

Changed content is indicated in green colored text in the manuscript.

I thankfully found this round of editing a good chance of improving the paper.
